# Current Insights in the Repeat Breeder Cow Syndrome

**DOI:** 10.3390/ani13132187

**Published:** 2023-07-03

**Authors:** Carlos Carmelo Pérez-Marín, Luis Angel Quintela

**Affiliations:** 1Department of Animal Medicine and Surgery, Faculty of Veterinary Medicine, University of Cordoba, 14014 Cordoba, Spain; 2Unit of Reproduction and Obstetrics, Department of Animal Pathology, Faculty of Veterinary Medicine, Universidade de Santiago de Compostela, 27002 Lugo, Spain; luisangel.quintela@usc.es

**Keywords:** cattle, estrus, repeat breeding, reproduction, risk factors, subfertility

## Abstract

**Simple Summary:**

Repeat breeder cow (RBC) syndrome encompasses cows failing three or more times to become pregnant, but with the special characteristic that their estrous cycles have a normal duration. Farmers and technicians commonly feel incapable of addressing this reproductive issue, and after numerous unsuccessful attempts to conceive, it is often necessary to cull cows. Important economic losses are linked to RBC syndrome due to diagnosis, therapy, or reduced milk production, among others. This review compiles information about recent knowledge on the pathogenesis of RBC syndrome, describes treatments that could be efficient, analyzes the risk factors involved, and slightly evaluates their economic impact on farms. While it is difficult to resolve certain cases due to the multicausal nature of this syndrome, remedial therapy and management strategies are listed.

**Abstract:**

Cows can have difficulties becoming pregnant, and in certain cases, these reproductive failures do not have an evident cause. Furthermore, when these failures are repeated three or more times with estrous cycles of normal duration and in the absence of evident clinical signs, it is considered repeat breeder cow (RBC) syndrome. A substantive incidence of RBC syndrome has been reported all over the world, which severely affects the farm economy. This paper reviews those studies particularly focused on RBC syndrome from 2000 to 2023 but also includes consolidated information until this date. Hormonal imbalances, undetectable oviductal or uterine defects, or poor oocyte or embryo quality have been reported as causes of RBC syndrome, while subclinical endometritis has been considered a relevant causal agent. However, it is unresolved why this condition is recurrent in certain animals, despite the implementation of corrective management actions or treatments. Recent studies evaluate the putative role of certain genes, factors, hormones, or proteins in the pathogenesis of RBC syndrome. Numerous risk factors contribute to the appearance of this syndrome, and some of them could be mitigated to partially prevent this infertility, while others cannot be changed. Due to the complexity of this syndrome, it is important to increase knowledge about the mechanisms involved, develop new diagnostic tools to differentiate causal agents, and implement new treatments to restore fertility. There is consensus about the huge repercussions of this syndrome on farm profitability, but further studies are now needed to describe its economic impact.

## 1. Introduction

Nowadays, dairy and beef cattle are being developed under demanding production systems due to increased competitiveness in their markets, amid an uncertain economic world situation characterized by high food and fuel prices, as well as the urgent climate emergency on our planet. In this context, farms need to be highly efficient and profitable, and the identification of non-productive animals is a priority to prevent economic losses. For many years, researchers and technicians have focused on the Repeat Breeder Cow (RBC) syndrome, in which cows exhibit estrus signs at apparently normal intervals (i.e., estrous cycles between 17 and 25 days) but repeatedly fail to be pregnant after at least three attempts, despite the absence of apparent anatomical abnormalities or infectious diseases [1,2,3,4]. This article will describe the multifactorial etiology reported in RBCs and list the risk factors considered to be underlying this reproductive failure. While many diagnostic tools for identifying reproductive pathologies are available, it is not practical or feasible to employ all of them, as they can be expensive or technically unavailable. Consequently, cows may experience reproductive imbalances or diseases that are not correctly diagnosed. On some occasions, these subfertile cows may eventually conceive after three or more cycles, but the economic impact of delayed conception often necessitates culling the animals because they become unprofitable and, in the case of dairy cows, produce less milk than the cost of their maintenance.

This problem has been extensively studied in various countries and regions, and researchers widely agree on its significant impact on the farm economy. By reviewing different studies, the frequency of RBC syndrome has been reported as approximately 5% in Jordan [5], 9% in the UK [6], 10% in Sweden [3], 11% in Bangladesh [7], 14% in Japan [8], 22% in the United States [9], 23% in Australia [10] and India [11], 25% in Spain [12], 36% in Cuba [13] and Ethiopia [14], and even as high as 62% in Indonesia [14,15]. The vast variation between countries can be attributed to various management practices, including the aptitude, production system, number of cows per farm, or genetic merit (mainly based on milk yield), among others. Studies have shown higher percentages of RBC syndrome in tie-stall barns [15], where hormonal treatment for reproduction control was not used [14,15], under tropical conditions [11,13,15], or in farms with high culling rate [12], which could be associated with poor animal welfare [16]. This global perspective suggests that the incidence of RBC syndrome could potentially be reduced and controlled.

The majority of studies on RBC syndrome have focused on dairy cows because their reproductive management allows for better monitoring of animals, making it easier to track estrus signs and insemination records. In contrast, beef cattle are typically raised under extensive systems, which makes obtaining such records challenging. However, the implementation of new precision livestock farming technologies, such as GPS collars, accelerometers, or proximity tags, among others, is expected to improve animal data collection and management. These advancements hold the potential to aid in the identification of subfertile cows, including RBCs, which is especially relevant in the case of extensive beef cattle farming due to the difficulty of obtaining reproductive records, as previously mentioned.

The subfertility observed in RBCs can be attributed to various causes, not only of maternal origin but also related to bull defects (as subfertility, anatomical defects of reproductive organs, low frozen–thawed semen quality, difficulties for mounting due to low libido or lameness, high hierarchy of infertile bulls) or management mistakes (wrong site or timing of semen deposition by technician, inadequate heat detection). Multiple factors have been reported as contributing to the occurrence of RBC syndrome, including subclinical endometritis [17], nutritional deficiencies [18], abnormal heat behavior or improper heat detection [19,20], mismanagement in artificial insemination (AI) [21], or endocrine dysfunctions [20,22]. In relation to hormonal alterations, factors such as elevated progesterone levels, abnormal follicular dynamics, delayed ovulation, and lower oocyte quality have been identified as responsible for subfertility in repeat breeder heifers [23]. Additionally, these causes are influenced by various risk factors that contribute to the occurrence of imbalances and the incidence of RBC syndrome, including factors such as age, parity, body condition, milk yield, environmental conditions, and peri- and postpartum imbalances, among others.

Successful reproductive performance in cows relies on the delicate interplay of hormone patterns, ovarian dynamics, estrus behavior, uterine functions, and mating or AI [23]. Several endogenous factors operating at the ovary, oviduct, or uterus can affect oocyte and embryo quality, thereby influencing the incidence of repeat breeding in dairy cows.

The current review on RBC syndrome focuses on analyzing the main participants that contribute to the occurrence of repeated estrus in cattle. Novel tools are being utilized to understand why these cows are unable to overcome subfertility situations and to identify the common factors among this population.

## 2. Literature Selection and Evaluation

This review utilizes comprehensive and up-to-date information on RBC syndrome, encompassing significant scientific advancements and findings from recent years. PRISMA guidelines [24] were used to ensure a systematic approach. To conduct this study, relevant information was sourced from databases such as Medline and Science Direct (last accessed on 9 March 2023); additionally, Google Scholar was searched to identify any additional scientific papers. The search queries employed were (“repeat breeder cow”), (“repeat breeding cow”), (“repeat breeder” AND (“cow” OR “heifer”)), and (“repeat breeding” AND (“cow” OR “heifer”)). These queries were applied to the title, keywords, and abstract sections of the papers. Full-text reports in the form of reviews or research articles written in English language were included, while conference or congress contributions were excluded. Initially, a total of 1403 papers were obtained from the mentioned search engines, and an additional 12 papers from other sources were reviewed. Studies within the timeframe of 2000 to 2023 were selected, and relevant information was extracted for analysis (Figure 1).

Selected papers were divided according to their scientific themes. They were assigned to “treatments for RBCs”, “subclinical and clinical endometritis”, “identification of genes, proteins, factors or hormones involved in the RBC syndrome”, “reproductive hormonal environment and quality of follicles and oocytes”, “risk factors in RBCs”, “nutrition and chemical components involved in RBC”, “anatomical defects in the reproductive tract” and “economic impact of RBC” (Figure 2).

## 3. Importance of Subclinical Endometritis in RBC Syndrome

Numerous studies have examined the impact of subclinical endometritis (SCE) on the occurrence of RBC syndrome, as can be seen when the most recent research literature is examined. This pathological condition may go unnoticed by farmers and technicians, but when a preventive program is established as a routine on a farm, SCE is often diagnosed in these cows. The literature established different thresholds of PMN for the diagnosis of SCE in cows [25]. Cytobrush and low-volume flushing are used for the diagnosis of SCE, but it is important to consider that the proportion of PMN decreases over the postpartum period. A fixed threshold of 18% PMN is established for samples taken between 20–33 days postpartum, 10% PMN at 34–47 days postpartum [26], and 5% PMN for cows between 21–62 days postpartum [27]. In contrast, for nulliparous heifers, the threshold for SCE is only 1% PMN [28]. Overall, many studies support the use of 5% PMN as a diagnostic threshold for SCE. Cows exhibit a weak post-mating inflammatory response after breeding, which differs from other species [29]. It is hypothesized that a certain influx of PMN into the uterus immediately or within the first week after AI might be associated with physiological and beneficial effects on conception [30,31].

The uterus serves as the environment in which the embryo should develop, relying on physical accommodation and hormonal support, and various factors, proteins, and genes are required for the embryo–maternal interplay. Embryonic losses frequently occur during the pre-attachment period, when the embryo is elongated and the trophectoderm is adhered to the uterine luminal epithelium, involving different endometrial cells. The malfunction of any of these mechanisms impedes reproductive success. In particular, uterine inflammatory conditions exhibit high endometrial expression of mucin 1 (MUC1), as well as IGF1 and IGFBP2 [32]. The role of MUC1 in endometrial content has been emphasized in preventing bacterial invasion in humans. Consequently, its role in the RBC has been investigated by evaluating the mRNA expression of MUC1 and cytokine genes in the endometrium. It is suggested that MUC1 might act as an endometrial anti-adhesive molecule, and during implantation [33]. It needs to be locally removed after day 15 of pregnancy, allowing other glycoproteins to stimulate trophoblast adhesion to the endometrium. Then, high expression of MUC1 could be linked to reproductive failure due to its prevention or delay of embryonic attachment. Progesterone, a pregnancy-supporting hormone, can also be involved because it induces the downregulation of MUC1 and indirectly acts on the endometrium by inducing various epithelial genes during the pre-implantation period [34]. A significant percentage of RBCs are diagnosed as SCE by uterine cytological sampling [17], showing altered endometrium, which not only promotes MUC1 expression, but also increases the levels of certain cytokines. It is suggested that those RBCs without SCE might have compromised their ability to downregulate MUC1 [35]. As a result, low-quality embryos may fail to implant due to the anti-adhesive properties of their glycocalyx. Various studies have investigated the inflammatory events of endometrial cells in RBCs, which exhibited higher mRNA expression of pro-inflammatory factors such as PTGS2 and CXCL3 [35]. Apart from that, preliminary studies conducted on RBCs with subclinical endometritis identified the presence of macrophages (CD14) in the endometrial stratum compactum and stratum spongiosum, suggesting their involvement in RBCs associated with embryonic mortality [36].

The postpartum period is widely recognized as influential for future reproductive outcomes in cows, and uterine disorders are often associated with inappropriate management during this period. RBC syndrome is considered one of the most important reproductive disorders following postpartum [8]. Cytological assessment of the uterus has revealed the significance of subclinical endometritis as a causal factor in RBCs, affecting up to 53% of them [17]. In contrast, other researchers have concluded that subclinical endometritis, uterine infections, or ovarian inactivity are not strongly associated with RBCs [37]. Poor quality of day 7 embryos and the presence of SCE have been shown to negatively influence early embryo development between days 7 and 16 of gestation, likely due to dysregulated embryo–maternal interaction resulting from lower progesterone levels, leading to conceptus loss in a sub-optimal uterine environment [38].

Ovarian steroid hormones stimulate the local production of growth factors and their receptors and regulate uterine function. Recently, epidermal growth factor (EGF) has received special attention as one of the most important components in regulating uterine function and embryonic development. Altered EGF production in the endometrium has been observed in RBCs compared to normal cows [39], suggesting that abnormal EGF levels contribute to uterine dysfunction. While RBCs are typically diagnosed using ultrasonography, hormonal assays, endometrial cytology, or oviductal patency assessment [40], there are few therapeutic choices for infertile cows if no particular etiology is discovered through these procedures. Strategies proposed in such cases include optimizing the timing of insemination based on fixed-time artificial insemination (FTAI) or treating endometritis through antibiotic infusion, antiseptics, or uterine lavage. Determining EGF levels in the endometrium between days 3 and 14 could be useful for diagnosing RBCs, as normalization of EGF profiles following treatment has been observed to restore fertility in RBCs [41]. Measurement of EGF has been proposed to identify RBC syndrome [39], while circulating steroid hormone levels (as estradiol-17ß or progesterone) are not specific or sensitive enough to diagnose this syndrome. In such cases, a uterine biopsy is required to diagnose uterine dysfunction caused by an altered endocrine environment. These early variations are suggested to cause asynchrony between the uterus and the blastocyst upon its entry into the uterus. Changes on day 14 could be associated with defects in embryonic survival and development. Growth factors located in the endometrial tissue could be considered indicators of endocrine environment disorders and may be involved in the RBC syndrome [39].

Recent studies have described the use of milk osteopontin (OPN) to normalize EGF levels in RBCs when infused into the vagina, resulting in a doubled conception rate compared to the control group [42]. Infused substances act on the regulation of uterine function or environment through specific pathways in cattle. It is important to note that this treatment is not suitable for those RBCs with ultrasonographic defects, cytologically diagnosed uterine diseases, or defects in oviductal patency. It could be hypothesized that OPN sensitizes immune cells in the vagina, leading to uterus-targeted or systemic changes through immune cell activation in the attached lymph nodes. OPN also plays a crucial role in the crosstalk between innate and adaptive immunity through the Th1/Th2 balance and macrophage immune responses [43].

As mentioned earlier, subclinical endometritis (SCE) has been considered a consistent etiology of RBC syndrome. Uterine contractility is an essential mechanism for facilitating the progression of spermatozoa after insemination or eliminating abnormal fluid and overcoming inflammatory changes. When contractility is compromised, fertility can be reduced.

The accumulation of fluid in the uterine lumen indicates difficulty in drainage, which can interfere with embryo implantation. However, studies conducted in RBCs indicate that the presence of uterine fluid (≥2 mm measured by ultrasonography) is not related to the diagnosis of SCE but reduces the probability of pregnancy by up to twofold. This observation underscores the importance of ultrasonographic scanning of the reproductive tract in predicting reproductive performance in RBCs [44].

## 4. New OMICs and Technologies to Understand the RBC Syndrome

The development of new analytical tools, such as OMICs, has opened up new insights into understanding the RBC syndrome. Certain genes’ participation, altered gene expression, and metabolomic changes mark new pathways that affect fertility in cows, suggesting new strategies to combat the RBC syndrome or minimize its impact.

The study of bacterial communities in the vagina has revealed the potential for evaluating the vaginal microbiome for designing therapeutic interventions in RBC syndrome [45]. Proteobacteria were found in subfertile cows, associated with dysbiosis during the postpartum period, and an abundance of Porphyromonas and Pasteurellaceae has been found in RBC syndrome [45], although further studies should be conducted to clarify their role in this syndrome. E. coli, another common bacterium, is also involved in the RBC syndrome, affecting the uterus function and causing ovarian dysfunctions with a negative influence on LH secretion and the CL lifespan [46]. In sum, inadequate vaginal and uterine microbiota can reduce uterine health and negatively affect fertility [47].

Metabolomic profiling in blood plasma has been used to identify metabolites that could help diagnose RBC syndrome [48,49]. Researchers reported lower levels of bile acids and kynurenine, and a total of 17 metabolites in the blood plasma of RBCs that were not present in control cows [49]. Imbalances of ketone bodies, such as acetoacetate (AcAc) and ß-hydroxybutyrate (BHB), have been associated with abnormal inflammatory conditions in RBCs throughout the activation of the NLRP3 inflammasome, which regulates sterile inflammation and IL-1β secretion [48]. Inflammation plays a role in the physiology and pathophysiology of pregnancy, and any disruption in immune function could lead to infertility, spontaneous abortion, and recurrent pregnancy loss [50,51]. Furthermore, the mRNA expression of inflammatory indicators, including IL-1β, in peripheral leukocytes was found to be higher in pregnant cows with early embryonic losses than in cows with maintained pregnancies [52]. In a similar syndrome observed in humans, in recurrent pregnancy loss, both IL-1β and NLRP3 inflammasomes are expressed at higher levels in endometrial human tissues [53]. Then, NLRP3 inflammasome activation and IL-1β overproduction in uterine tissue might be proposed as tentative causes of RBC syndrome [48]. According to the reviewed literature, an inflammatory response could be associated with RBC syndrome, and perhaps anti-inflammatory therapy might partially resolve this syndrome.

Gene expression has also been evaluated to identify new etiological causes in RBC syndrome. Puglisi et al. (2013) described a total of nine genes expressed in cumulus-oocyte complexes (COCs) from RBCs, compared to control cows [54]. In RBCs, lower expression of annexin A1 was observed, which affects the in vitro maturation of oocytes and upregulates the phospholipase A2 gene. There was also reduced expression of the lactoferrin gene, which is linked to lower in vitro fertilization rates and lower quality embryos. Under-expression of interferon-stimulated exonuclease 20 kDa was found in RBCs, compromising reproductive behavior and the response of the COC to estrogens. Moreover, the oxidized low density lipoprotein receptor 1 was under-expressed, which is associated with ovulation defects. The fatty acid desaturase 2 gene was over-expressed in RBCs, affecting the metabolism of polyunsaturated fatty acids as well as oocyte growth and differentiation. Glutathione S-transferase A2 was over-expressed, while glutathione S-transferase A4 genes were down-regulated, both of which are implicated in oxidative damage and oocyte viability. Similarly, the glutathione peroxidase 1 gene was up-regulated and associated with increased oxidative stress. Additionally, endothelin receptor type A was increased in the COC of RBCs, suggesting dysregulation of meiotic resumption and follicle rupture, among other actions [54].

Using gene ontology, it has been determined that interferon-τ-stimulated genes (as ISG15, SLC2A1, SLC27 A6, and CXCL), but also peroxisome proliferator-activated receptors, are involved in coding the trophectoderm cell proliferation and migration and prompted conceptus elongation on day 16 [55]. The interruption in crosstalk between the endometrium and conceptus may impair conceptus elongation in RBCs with SCE. The implementation of genomic selection improves breeding programs in cattle but might also partially reduce the incidence of RBC syndrome.

Another area of interest in understanding the RBC syndrome is the environment in which the oocyte develops. The follicular fluid from RBCs induces disadvantaged development for oocytes compared to fluid obtained from fertile virgin heifers, reducing in vitro maturation, fertilization, and blastocyst yield [22]. Proteomic and metabolomic techniques have revealed differences in the quantity of proteins and metabolites identified in the preovulatory follicular fluid in RBCs and normal cows, which are related to follicular function and oocyte competence [56,57]. Induction of the steroidogenic shift in the ovulatory follicles and expression of LH surge-dependent genes in granulosa cells relies upon the time of the occurrence of the ovulatory LH surge and follicular fluid composition [58]. Therefore, abnormalities in the LH surge may provide an inappropriate microenvironment for the oocyte. Lower cumulus cell expansion and nuclear maturation were observed when RBC fluid was used, suggesting an inadequate environment for oocytes [22].

Inflammatory processes associated with endometritis or mastitis result in a high presence of lipopolysaccharides in the follicular fluid [59], causing ovarian dysfunction and poor oocyte developmental competence. Altered granulosa cell gene expression and lower follicular estradiol production are linked to these dysfunctions [60]. Follicular fluid secreted by the granulosa cells contains soluble factors such as estradiol, progesterone, IGF1, TNFα, IL6, and stem cell factors, all of which influence oocyte development and competence [61,62]. Proteomic evaluation of the follicular fluid has identified eight proteins that significantly differ between fertile and RBCs, interfering with follicular function and oocyte quality. Low or null levels of α-1-antiproteinase (SERPINE1), heparan sulfate proteoglycan core protein (HSPG2), IL-1 receptor accessory protein (IL1RAP), prothrombin (F2), and collagen alpha-2(I) chain (COL1A2) were detected in the follicular fluid of RBCs; in contrast, high levels of the other proteins, inter-α-trypsin inhibitor heavy chain H1 (ITIH1), complement component C8 alpha chain (C8A), and tissue inhibitor of metalloproteinase 2 (TIMP2), were reported [63]. They are associated with follicular metabolism, immune status, and regulation of COC matrix proteins, affecting fertility by different pathways [20,63,64,65,66].

## 5. Anatomical Defects Associated with RBC Syndrome

Oviductal defects, such as tubal stenosis or occlusion, have been associated with the RBC syndrome [67]. These conditions prevent the movement of oocytes and spermatozoa through the oviducts, and if fertilization occurs, the passage of the embryo into the uterus will be difficult. In cows with oviductal alterations, their estrous cycle remains unaffected, showing a normal duration. Sometimes, these issues can be resolved spontaneously. Confirming oviductal patency is challenging, but various methods have been used in studies, including examination at slaughterhouses [68], ultrasonographic contrast liquid [69,70], and dyes (phenol-sulphon-phthalein) [67] or granules (starch) [68]. Tubal inflammation is commonly associated with endometritis, and it reduces sperm motility and impairs sperm–oviduct interaction. The transport of early embryos can also be compromised due to mucus accumulation and reduced ciliary beat frequency in the oviduct. Research has indicated that around 20% of RBCs show bilateral oviduct occlusion or stenosis, while unilateral defects are reported in 24% of cows [67], highlighting the significance of these defects in the etiology of RBC syndrome.

In the search for other causes of RBC syndrome, studies analyze cervical consistency and patency [71]. It is reported that RBCs with normal uterine findings may exhibit cervicitis, which is independent of endometritis [71]. This suggests the importance of conducting a comprehensive breeding soundness examination that includes evaluating the external genitalia, perineal/vulvar conformation, and internal genitalia via vaginoscopy in RBC cases. Cervicitis is characterized by a prolapsed and swollen portio vaginalis cervicis with reddening and can negatively impact conception rates and the number of pregnant cows at 200 days into lactation [71]. Additionally, identifying abnormal vulvar conformation in cows is crucial, as it is associated with pneumovagina, vaginitis, or endometritis (or SCE). According to the definition of RBC syndrome, when reproductive inflammatory alterations are present in cows, they should not be considered RBCs [72]; however, certain inflammatory disorders cannot be correctly diagnosed. In other instances, cows may experience SCE, which negatively affects reproductive performance and can be included in the definition of RBC syndrome.

## 6. Nutrition

The multifactorial etiology of RBC syndrome also includes causes related to nutrition or metabolism. Although the exact pathways are not fully understood, studies have examined the impact of nutritional imbalances on hormones, oocytes, or embryos, among others. Obesity or malnutrition can lead to reproductive dysfunction and subfertility due to inadequate metabolic function, which is closely linked to adipose stores and neuroendocrine function [73,74].

In general, the literature compiles numerous studies analyzing serum biochemical elements in RBCs. It has been established that RBCs have lower levels of total protein, total cholesterol, and glucose compared to fertile cows [74,75], but higher levels of urea nitrogen [75]. Others observed higher values of gamma-glutamyltransferase in RBCs, which is an indicator of hepatic abnormalities, suggesting the importance of liver functioning in relation to RBC appearance [76]. In contrast to other studies, various analyzed parameters such as alanine aminotransferase, aspartate transaminase, alkaline phosphatase, lactate dehydrogenase, creatinine kinase, albumin, globulin, total bilirubin, blood urea nitrogen, creatinine, glucose, triglyceride, non-esterified fatty acids (NEFA), calcium, magnesium, and inorganic phosphate did not show any significant differences [76]. A previous study conducted before AI showed more controversial results than previously mentioned, noting significant variations in calcium, phosphorus, and magnesium between pregnant and open RBCs, but no differences were observed for glucose, copper, or zinc [77]. Generally, these studies involved blood sampling during lactation, and the cows were at different stages of lactation, which could have influenced the results. In conclusion, after reviewing all the papers concerning RBCs, it can be inferred that nutritional imbalances clearly affect reproductive performance and, therefore, are likely to be associated with RBC syndrome. However, it is not possible to identify specific chemical elements that characterize this syndrome.

Leptin is a protein produced by adipocytes and modulated by insulin, glucocorticoids, and sexual steroids. It provides information to the brain regarding the available energy reserves, and it also inhibits the production of hypothalamic neuropeptide, a powerful appetite-stimulating factor. A study conducted during the postpartum period reported lower concentrations of leptin in RBCs compared to fertile cows [74]. The strong association between fertility and nutrition suggests that leptin may play a crucial role in modulating reproductive function, particularly during the postpartum period, when negative energy balance and low leptin levels coexist. The resumption of ovarian activity will depend on appropriate nutritional and health management during the postpartum period [78]. Additionally, low leptin levels can induce fasting behavior, resulting in reduced food consumption, and may lead to altered ovarian function due to a reduction of LH receptors and estrogen synthesis by granulosa cells [79].

Nutrition can influence insulin levels and various growth factors, which in turn impact follicular development, steroidogenesis, oocyte maturation, and embryo development, thereby increasing the incidence of RBC syndrome. Based on this understanding, insulin has been suggested as an alternative therapy to alleviate reproductive failures in RBCs because it can increase intrafollicular and peripheral IGF-I levels [80]. A slight improvement has been observed in reproductive performance in RBCs treated with a long-acting form of insulin on days 8, 9, and 10 of the estrous cycle [81], but the beneficial effects of this treatment have not been clearly demonstrated.

Dietary supplementation with n-3 polyunsaturated fatty acids (n-3 PUFA), typically found in fish oil, has been recommended to reduce the incidence of RBC syndrome. Administering n-3 PUFA for two weeks before and after AI has been reported to enhance utero-ovarian functions and embryonic survival in postpartum dairy cows [82]. This supplementation has been associated with larger preovulatory follicle size, higher serum progesterone levels, and an increased abundance of mRNA of interferon-stimulated genes [82]. These findings further emphasize the importance of feed quality and highlight the significance of genomic tools for better understanding RBC syndrome.

Negative energy balance (NEB) occurs during the peripartum period, typically in the last week of pregnancy and the first two months after parturition. During this stage, cows show low serum concentrations of glucose and insulin, which stimulate the mobilization of adipose tissue. This results in an increase in non-esterified fatty acids (NEFA), serving as an energy source for these animals. However, sustained high levels of NEFA and ß-hydroxybutyrate can negatively impact reproductive performance, leading to anovulation, reduced pregnancy rates, and higher embryo loss [83], due to their detrimental effects on immunity and postpartum health [84]. In vitro studies have confirmed that NEFA has adverse effects on the synthesis of interferon-gamma and IgM, as well as on the functionality of polymorphonuclear leukocytes [85,86]. Additionally, deficiencies in antioxidants and vitamins (A and E), which play a role in the immune response of cows, have been associated with increased susceptibility to postpartum diseases [87]. Even the germinal cells, such as oocytes, are negatively affected by NEB, despite the existing regulatory mechanisms within the intrafollicular system [88]. Reports have indicated that NEB leads to reduced oocyte competence, apoptosis of cumulus cells, impaired fertilization, and diminished blastocyst cleavage and development. Furthermore, NEB disrupts hormonal function, altering the normal synthesis of GnRH, LH, and other hormones [84].

The nutritional status and metabolism of specific nutrients play a critical role in ensuring adequate immune and other cell functions in dairy cows [89]. Nutritional deficiencies can affect the reproductive process in cows, especially in dairy cows with higher nutritional requirements. Analyzing the chemical content in blood plasma is not practical, so it is more reasonable to monitor the rations, body condition, or weight to minimize their negative impact on reproduction.

## 7. Steroid Hormones, Biochemical Components, and Gametes/Embryo Competence in RBC

Alterations in steroid hormone production and release are observed in RBCs and encompass various defects that are challenging to identify at the farm level. Irregular progesterone profiles are associated with temporal endocrine imbalances, abnormal ovulation, fertility failures, or early embryo death [4,9]. These imbalances can affect the oocytes, the spermatozoa, the embryos, and the reproductive behavior of affected cows [90]. Early embryo loss can be derived from the aforementioned alterations. In general, RBCs have higher levels of estradiol and progesterone around ovulation, which could promote luteal dysfunction and asynchrony between the embryos and the uterine environment and development [91]. Elevated progesterone levels before ovulation in cows can lead to delayed, irregular, or absent ovulation. Additionally, the presence of persistent follicles, characterized by prolonged and/or high estradiol levels, can contribute to the development of subfunctional CL; as a result, insufficient progesterone is produced, which impedes the maintenance of pregnancy [4]. Moreover, ovarian cysts have been observed in RBCs, potentially as a result of prior hormonal or uterine disturbances [92].

Regarding RBC syndrome due to subfunctional CL, low progesterone levels during the first week after ovulation make it difficult for the development and implantation of the embryo, leading to early embryo death. Additionally, certain factors or proteins associated with pregnancy may be diminished, such as tau interferon, a trophoblastic substance involved in early pregnancy recognition and related to progesterone levels. Progesterone levels lower than 2 ng/mL until day 8 post-ovulation indicate low luteal quality and increase the likelihood of RBC syndrome [4]. Furthermore, it is suggested that RBCs show overproduction of ß-endorphins and free radicals (such as ROS) due to stressful situations, which negatively affect the functionality of CL [93].

Adequate progesterone levels during the preovulatory phase are also important [20]. The presence of suprabasal levels (usually resulting from partial luteolysis failure) can prolong the growth period of the ovulatory follicle and promote the overmaturation of the oocyte, thereby reducing its fertilizing capacity. Consequently, there may be observed asynchrony between AI and ovulation (which it is delayed or impeded), hormonal imbalances (elevated estradiol), formation of a subdued CL, and endometrial changes [90].

It is reported that RBCs exhibit more intense behavioral estrus signs and higher estradiol levels compared to fertile cows, although the follicular diameter is comparable in both groups [20]. This suggests increased steroidogenic activity in RBCs, which could negatively affect oocyte quality and development. RBCs also exhibit a short proestrus period and a subdued LH secretory pattern before the LH peak [20].

The etiology of RBC syndrome is partially linked to folliculogenesis failures, which affect oocyte competence and reduce the chances of fertilization and embryo vitality and development [20]. Studies have noted higher estradiol concentrations in the follicular fluid of RBCs, but no significant variation in androstenedione and progesterone levels in the same fluid. Following in vitro embryo production in RBCs, which involved ovum pick-up for collecting oocytes, no differences were observed in the number of follicles aspired per session, oocyte recovery rate, or cleavage rate, although the production of blastocysts was markedly reduced [92]. More recently, a study describes a significantly higher percentage of RBCs when serum anti-Müllerian hormone (AMH) concentrations are lower [94]. This hormone is associated with the antral follicle count and informs about the size of the ovarian primordial follicle reserve pool [95]. Therefore, the reduced fertility in cows with low AMH concentrations could be attributable to lower oocyte quality, reduced sensitivity of theca, granulosa, and luteal cells to FSH and LH, impaired embryonic developmental competence, and a relatively lower progesterone concentration during the estrous cycle. Interestingly, higher AMH levels have also been associated with suboptimal fertility, suggesting that a larger antral follicle count causes an imbalance in gonadotropin secretion, with greater LH and lower FSH secretions [94]. Monitoring AMH values is suggested to reduce future RBC syndrome incidence, as it is a moderately heritable trait, although further studies should be conducted.

Several studies have been conducted to identify RBCs based on circulating levels of different biochemical components. Although differences have been described between RBCs and normal cows for gamma glutamyl-transferase, total protein, total cholesterol, glucose, or urea nitrogen [75,76], these parameters are not accurate enough for a specific diagnosis. Many different deficiencies or pathologies can modify biochemical parameters in cows, making it a less specific diagnostic method for this syndrome. Nonetheless, deficiencies in various chemical parameters such as calcium, phosphorus, or magnesium have been associated with the abnormal metabolic or physiological environment in RBCs [77], as mentioned in the previous section. The multicausal etiology of RBC syndrome makes difficult the use of these parameters as diagnostic tools.

As mentioned, hormonal imbalances, such as luteal insufficiency, have been strongly suggested as a cause of RBC syndrome, which is associated with embryonic mortality. Low concentrations of progesterone during the first 8 days after estrus are associated with RBC syndrome. Studies examining the effects of progesterone supplementation after AI have shown a positive effect when it is administered between days 5 and 19 post-AI [96]. This covers the critical period of early pregnancy during the first week after AI, enhances embryo growth, and ensures high progesterone concentrations until the attachment of the chorion and endometrium.

Since defects associated with fertilization and oocyte quality have been linked to RBC syndrome, procedures to bypass these problems (related to the ovarian environment, oocyte development, ovulation, or oviductal embryo transport, among others) have been suggested, such as the embryo transfer procedure [97,98,99]. Furthermore, it should be considered that the causative agent of subfertility in RBC is not always known, and then, the probability that the transferred embryo was successful is very uncertain. For this reason, so-called “therapeutic embryos”, obtained by in vitro production from oocytes derived from slaughterhouse ovaries, are transferred to RBCs, avoiding the use of genetically valuable embryos, which are more expensive.

Genetic defects have also been mentioned as a potential cause of RBC syndrome, including chromosomal or genetic abnormalities, often related to inbreeding or aged gametes. However, there is limited evidence regarding the exact genetic mechanisms or defects associated with RBC. These defects can be observed in oocytes, embryos, and the genital tract. One study reported 1/29 translocations and sex chromosome aberrations in 14.3% of RBC cases [100], which should be considered a significant but relatively small factor involved in this syndrome.

## 8. Therapeutic Control of the Estrous Cycle

Some RBCs can achieve pregnancy in subsequent estrous cycles, even in the absence of any treatments. If any therapy is implemented (antibiotics, hormones, mineral supplementation, uterine antiseptics, or others), reports indicate that only 21% of these subfertile cows will successfully conceive [11]. Approximately 24.3% to 31.4% of RBCs will be pregnant within 210 days after calving [4,8], but the likelihood of RBCs becoming pregnant after 300 days postpartum is extremely low, with only an additional 8% chance [8]. Table 1 shows a compilation of RBC treatments published in recent scientific papers.

Hormonal imbalances and asynchronic reproductive events have been reported in RBCs, such as delayed LH surge, subluteal progesterone during diestrus, etc. Therefore, controlling the estrous cycle and ovulatory events has been suggested as a recommended treatment. Therapeutic protocols for FTAI based on the synchronization of ovulation (such as Ovsynch) or pre-synchronization (such as Double Ovsynch or G6G) have become popular for achieving high fertility in the first service postpartum. These treatments can improve and synchronize ovulatory events, such as promoting proper follicular growth, facilitating ovulation, and supporting corpus luteum (CL) formation. For these reasons, when used in RBCs, the mentioned treatments can improve the reproductive events, achieving conception rates of around 50–60% [101,102], although further studies should be carried out. Additionally, other protocols based on progesterone supplementation have been effective [96,103]. Alnimer and Husein (2007) described a treatment based on estradiol, PGF2α and GnRH, previously primed by progesterone, which resulted in improved estrus detection and elevation of plasma progesterone levels [104]. A combination of GnRH, progesterone, and meloxicam has offered higher conception and pregnancy rates in RBCs (over 10%) [105]. Treatments based on a 7-day FTAI protocol overcame the delayed ovulation and anovulation described in RBCs [90], obtaining a higher preovulatory follicle size and pregnancy rate [106,107]. It is interesting to highlight that those cows with CL (with a diameter higher than 15 mm) at the initiation of the treatment achieve higher fertility [106,108]. Factors such as high peak yields, high milk protein, increased parity, increased lactation period, high temperature-humidity index, or short dry periods negatively impact the fertility obtained after these treatments [108].

Another treatment designed to overcome RBC syndrome is based on HCG (1500 IU IM) on days 4 and 6 after TAI [109,110,111]. This treatment aims to reduce embryo deaths occurring around days 6–8 after AI by increasing tau interferon [112] and/or raising progesterone levels through a positive effect on the primary CL or inducing the formation of accessory CL [113]. In addition, it is suggested that monitoring follicle size and the vascularity of the largest follicle and uterine artery at the beginning of treatments could predict ovarian response in RBCs, with better outcomes when vascularization is higher [114].

The timing of insemination is very important, as demonstrated when a higher overall pregnancy rate is obtained in RBCs using double AI and GnRH administration [115]. Many studies report a positive effect of GnRH at AI, concluding that it could improve the pregnancy rate in a dose-dependent manner due to its action on progesterone release and ovulation timing in RBCs [116]. In general, it is estimated that the pregnancy rate could increase between 18% and 50% [117], suggesting that ovulatory disturbances are frequently the cause of the RBC syndrome [116]. In contrast, other studies do not report positive effects [109,118]. Recently, the use of single and double doses of dephereline, an analogous GnRH, has been explored to treat RBCs on days 5–7 post-AI [119]. A double dose (250 µg) increased the pregnancy rate and suggested that it could improve embryo survival. There is evidence that the administration of GnRH agonists in RBCs between 7 and 14 days after AI stimulates the development of a second CL and promotes embryo survival, resulting in a significant increase in pregnancy rate (11 percentage points compared to the control group) [120].

Using high-producing lactating RBCs as recipients in an embryo transfer (ET) program reports poor results. It is considered that they have an increased metabolism of estradiol and progesterone, compromising estrus expression and pregnancy establishment [121,122]. To overcome these difficulties, the use of hormonal protocols for estrus induction has been proposed. Higher conception rates are obtained in RBCs through ET compared to AI (41.7% vs. 17.9%), mainly during the summer [123].

RBCs diagnosed as SCE showed higher pregnancy rates when treated with antibiotics (after carrying out an antibiogram). The use of gentamicin (4 mg/kg IM + 200–400 mg in 16 mL saline in the uterus) for 5 days showed better results than enrofloxacin (in a similar dosage) in these RBCs [124].

As mentioned before, defects in the uterine contractibility provoke reproductive failure and RBC syndrome. Scopolamine, a parasympatholytic drug, has been proposed as a treatment to modulate uterine contractility and increase the conception rate in RBC cases with hormonal alterations or contractile and biochemical deficits. A recent study reported a significant increase in conception rates in RBCs treated with scopolamine (80%) compared to non-treated RBCs (25%) when administered on the day of estrus (12 h before AI) [125].

## 9. Risk Factors Associated with RBC

There are many factors influencing the incidence of RBC syndrome, but researchers have used different statistical approaches (i.e., odd ratio, chi square, mean comparison) to evaluate their effects (Figure 3, Table 2). The following section describes the risk factors that have been more frequently reported in the literature.

Age: As age advanced, higher probability to become RBCs was described in many studies [6,7,10,14,126]. It has been attributed to poor oocytes quality due to endocrine disturbances and depletion of the ovarian reserve [127]. Elder females usually show deficiencies in hypothalamic or pituitary hormonal release, as can be noted when LH or FSH levels are measured, and there is also a decline in their ovarian response capacity. The effect of age could be associated with cumulative disorders or pathologies over time, both of which affect reproductive functions. However, the age at which cows usually are culled, i.e., around 6 years, is not associated with infertility, as occurs in older cows. In general, a higher incidence of RBC syndrome is attributed to older cows, not specifically due to age itself but rather derived from postpartum diseases such as milk fever, dystocia, or retained placenta [10]. Table 2 shows that all the studies conducted on RBCs to evaluate the effect of age of cows on the incidence of RBC syndrome were significant, noting that the higher the age of cows, the higher incidence of this syndrome.

Parity: A higher prevalence of RBC syndrome has been observed in cows with three or more parities [7,8,14,128,129]. Each parturition is a risk period for the cow, and reproductive activity could be altered by calving difficulties (dystocia, cesarean, retained fetal membranes) or postpartum diseases (inadequate uterine involution, inflammatory affectation), promoting reproductive failures in subsequent estrous cycles in cows with more parities. And as mentioned before, periparturition is one of the most critical periods for immunosuppression in cows. Recently, a trial conducted on beef RBCs revealed that cows at low parity with repeated cycles have a higher probability of failing to become pregnant again in subsequent parities in comparison to cows with one or two repeated cycles. Furthermore, cows that require four or more AIs to achieve pregnancy will have greater challenges in becoming pregnant in successive parities [129]. However, some studies conducted in RBCs did not observe significant effect associated with parity [126,130].

Environmental conditions: The effect of heat stress and humidity on the RBC has been described. Heat stress is considered a causal factor for reduced fertility in cows, and consequently, it also promotes the RBC syndrome [131]. High temperatures modify normal ovarian follicular dynamics, alter the dominance linked to the dominant follicle, induce follicular codominance, decrease the length of the estrous cycle, or diminish the competence of oocytes to become blastocysts [132,133]. Due to all these negative consequences, embryo transfer has been proposed as a procedure to mitigate them [131]. During summer, RBCs show higher cutaneous temperature, respiration rate, and rectal temperature compared to heifers (with lower thermoregulation capacity). Higher fragmentation of blastocysts was observed in RBCs during the summer, suggesting their greater sensitivity to heat stress compared to other cows [131]. A hot environment has also been linked to a reduced number of copies of mitochondrial DNA and increased nuclear-encoded transcript related to mDNA transcription and replication and induces apoptosis in RBC oocytes [134]. It is expected that indigenous animals suffer few reproductive imbalances due to their lower milk yield requirements, lower use of AI (resulting in minimized semen quality and problems associated with ovulation and heat detection), and better adaptation to climatic conditions, which reduce the appearance of certain diseases and, of course, reproductive failures [11]. Then, the effect of breed genotype has been analyzed since local breeds, which are better adapted to the climate and natural conditions, show a lower incidence of RBC syndrome [128,135].

Peripartum abnormalities: Parturition is an important event for cows which suffer sudden physiological modifications and need to be adapted to this new situation. Periparturient hormonal changes have a negative impact on immune cell function [89]. In lactating dairy cows, hormonal and metabolic changes occur approximately three weeks before parturition as a result of negative energy balance (NEB), which can affect the function of lymphocytes and neutrophils, contributing to postpartum immunosuppression. As a result, cows are particularly susceptible to infections, both clinical and subclinical, during the transition period. Pathologies such as metritis, dystocia, retained fetal membranes, and metabolic imbalances are frequently observed and are associated with RBC syndrome [126,130,136]. The RBC condition is considered repeatable in subsequent lactations [129,137,138], and then it is very important to reduce the risk factors to diminish its incidence. Delayed heat after calving could be prone to RBC syndrome, perhaps due to prolonged periods of NEB [10].

Body condition: Cows with medium (2.75–3.5) or poor (<2.5) body condition scores showed a higher prevalence of subfertility compared to those with a higher score (>3.75) [14]. NEB in those cows with lower body condition scores reduces hormonal release, appearing fertilization failures, and early embryonic deaths [139]. NEB is reflected in body weight loss, which is associated with low levels of protein, minerals, vitamins, and/or energy [140].

Milk yield: As mentioned before, the condition of high genetic merit in dairy cows is associated with a greater risk of reproductive failure [140]. High nutritional demand and high metabolic effort are exerted by the cows when they produce a high quantity of milk, and a negative effect on reproductive performance is observed [14]. The increasing days to peak milk production in primiparous cows are associated with RBC syndrome, perhaps as a consequence of stress due to nutrition, management, or diseases in their first lactation or because insemination coincides with the moment of higher milk yield and higher metabolic demands [10]. Higher presence of fat and protein in milk correlates with higher probability of conception failure, perhaps due to fat mobilization. High-producing lactating cows have an increased metabolism of estradiol and progesterone, and, as a result, estrus expression and pregnancy establishment are compromised [121,122].

Reproductive management: The use of AI is also associated with a higher prevalence of RBC syndrome compared to natural mating [14]. It could be associated with the technician, due to incorrect insemination technique (dirty practice, improper insemination practice, wrong timing for AI) or defective heat detection skills. In contrast, natural mating overcomes the majority of these problems, but other inconveniences, such as semen quality or venereal disease transmission, could still occur. Insemination during the early stages after parturition has also been considered a factor that increases the probability of RBC syndrome. It has been observed that RBCs inseminated twice in the heat or with a single AI using GnRH improved the conception rate by around 20 percentage points [7]. In relation to the sires used for natural mounting, there is no influence on the incidence of RBC syndrome [141].

Herd size and production system: A higher probability of expressing RBC syndrome was observed in extensive and semi-extensive production systems and in large or medium-sized herds [14]. Small herds allow for more meticulous heat detection, usually resulting in AI being carried out at the proper time, and farmers can attend to any problems, such as during parturition, more promptly, reducing sequelae that could increase the number of RBCs on the farm.

## 10. Economic Impact of RBC Syndrome

RBC syndrome is considered one of the most significant reproductive issues on dairy farms, alongside other problems such as anoestrus, retained placenta, dystocia, abortion, stillbirth, purulent vaginal discharge, and uterine prolapse. To estimate the cost of RBC syndrome, models take into account parameters such as veterinarian’s fees, extra semen costs, expenses for therapeutic drugs, labor costs, decreases in milk yield, reduced calf production, losses due to culling and replacement, losses resulting from extended calving intervals, and on-farm deaths [11,40,142].

A comprehensive economic study on RBC syndrome was conducted in Michigan, noting that the highest economic losses or costs were associated with milk production, culling, and the cost of additional semen, among other factors [143]. The authors suggested that proper nutrition should be provided during the dry period to prevent cases of milk fever or dystocia and reduce the incidence of RBC syndrome. According to studies, not only the postpartum and periovulatory periods are involved in RBC syndrome, but the dry period is also important in reducing estrus repetition in cows [144].

It is estimated that each open day costs around 5 €, and RBCs have a longer calving–conception interval (187 days) compared to healthy cows [8]. However, costs are also associated with the use of additional semen doses and insemination practices, veterinary treatments, increased culling and replacement costs, and loss of genetic gain due to longer generation intervals [6]. The majority of the net cost of RBCs (79%) is attributed to milk loss, while around 23% is due to culling [143]. The cost of RBCs increases when more inseminations fail due to longer open days, resulting in higher expenses, and involuntary culling is estimated to cost around USD 500–1000 per cow. To understand the importance of treating RBCs for dairy farm profitability, a profit of USD 30.2 per RBC treated with GnRH is reported [133].

In almost all cases, the diagnosis of RBC syndrome is complex, and the required methods for an accurate diagnosis may be too expensive to be carried out individually. This is the case for tubal patency diagnosis, where uterine flushing or echodense contrast agents are needed, or for hormonal determination, which requires serial sampling. Additionally, RBCs with high genetic merit could be involved in therapeutic embryo transfer or in vitro production programs. However, these animals often have compromised fertility, and the cost of these procedures may not be justified considering their lower reproductive success and their impact on farm profitability. Therefore, the diagnosis and treatment of RBCs should be carefully considered based on the causal agent.

In addition, RBC syndrome should also be discussed in light of current trends regarding animal welfare and environmental impact in order to determine the most appropriate productive lifespan for cows, taking into account factors beyond farm-level profitability [145]. Improving reproductive efficiency can provide opportunities to extend the productive lifespan of cows, thereby increasing profitability and improving societal acceptance of dairy production.

Failure to conceive is described as one of the main reasons for culling cows. Nowadays, the discussion goes beyond the economic aspects associated with infertility and also considers the environmental and animal welfare implications of open cows, which are factors influencing culling decisions, including the age at which animals should be culled. There is a growing societal pressure to protect animals, which raises questions about the appropriate age for culling cows. While the natural lifespan of a cow is around 15–20 years, dairy cows are typically kept on a farm for only 5–6 years due to economic considerations. However, this decision is not in line with the sustainability of the production system.

Early culling leads to a higher number of young animals (heifers and primiparous cows) on the farm, resulting in increased greenhouse gas emissions (more methane and phosphorus per unit of milk) [146]. In dairy cows, it should be noted that maximum milk production is usually achieved in the fifth lactation [147]. Therefore, early culling does not take full advantage of the milk production potential and is not environmentally friendly, as previously mentioned [145].

Determining the optimal economic productive lifespan in dairy cows requires considering three factors: rational economic decision-making, changes in cow performance over time, and the accelerated genetic progress associated with new reproductive technologies and genomic testing [145]. Additionally, there are many other variables that can influence decision making on farms, making it challenging to establish universal rules in this area.

While RBCs may have longer calving–conception intervals, leading to potential considerations for culling, milk production levels should also be taken into account. High-yielding dairy cows may need to be maintained for a longer period without economic loss compared to open cows with lower milk production [145]. Decisions to keep these cows on the farm for an extended period provide more opportunities for pregnancy, avoiding culling, maximizing milk potential, extending lifespan and welfare, reducing the need for replacement with heifers, and decreasing the environmental footprint. As mentioned, current responsible and sustainable production systems should not rely solely on economic decisions, and this debate should be addressed.

## 11. Conclusions

In general, preventive measures are preferable for addressing RBC syndrome due to the challenge of achieving an accurate diagnosis. Based on the available evidence, the following measures might be adopted to reduce the incidence of RBC syndrome.

(a)The herd should focus on reducing the occurrence of periparturient diseases and minimizing the depth and duration of NEB (or nutrient deficit) in recently calved cows.(b)Hormonal treatments for estrus and ovulation induction (such as GnRH, hCG, PGF2α, and/or progesterone) can help maintain a favorable hormonal environment in cows. This approach can overcome issues like anovulation, mistimed AI, inadequate LH production, or the formation of a subfunctional CL, among others.(c)The presence of CL at the beginning of the hormonal treatment increases pregnancy rates in RBCs compared to cows without functional luteal structures.(d)Reproductive management practices, such as AI skills, hygiene during AI, and semen quality, should be regularly checked and improved.(e)Implementing a protocol for diagnosing, treating, and preventing SCE is important, as it is considered a “silent cause” of RBC syndrome.(f)Nutrition plays a role in this syndrome, so it is recommended to monitor the body condition score (BCS) and/or body weight as indicators of nutritional status.(g)In vitro production and ET are valuable technologies for preserving the genetic merit of certain RBCs. However, the use of “therapeutic embryos” in RBC recipients should be carefully evaluated, as the profitability of this practice depends on whether these cows are suitable recipients.(h)If RBCs fail to conceive around 300 days after parturition, the likelihood of resolving the syndrome becomes significantly low, and culling should be considered.

Despite advancements in understanding the involvement of certain genes, proteins, and factors in RBC syndrome, few therapeutic advances have been reported for reducing this syndrome. New developments based on translational medicine may offer opportunities to better comprehend and address RBC syndrome, thereby helping mitigate its economic impact on cattle farms.

## Figures and Tables

**Figure 1 animals-13-02187-f001:**
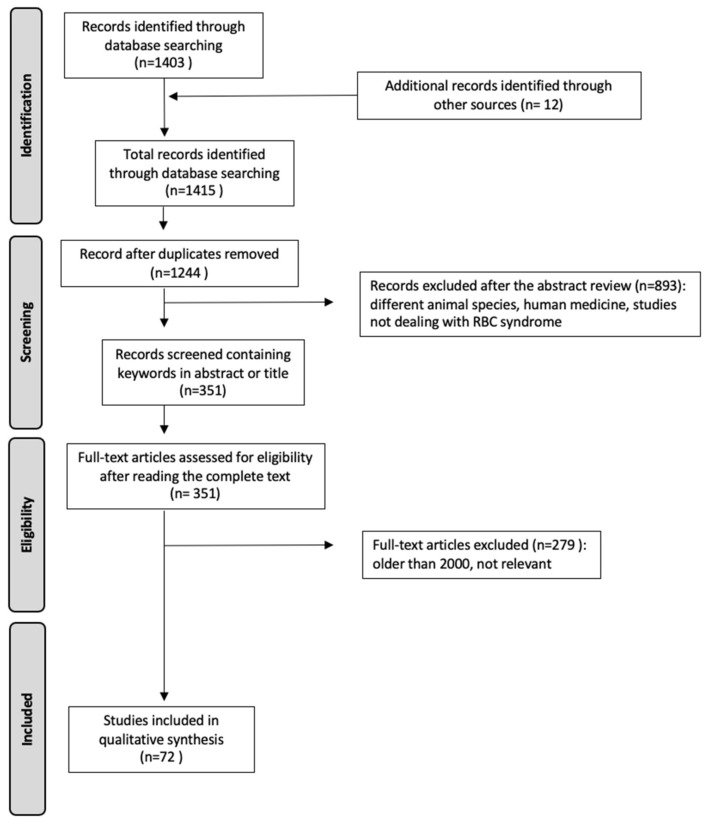
The flowchart illustrates the process followed to identify, exclude, and select the articles.

**Figure 2 animals-13-02187-f002:**
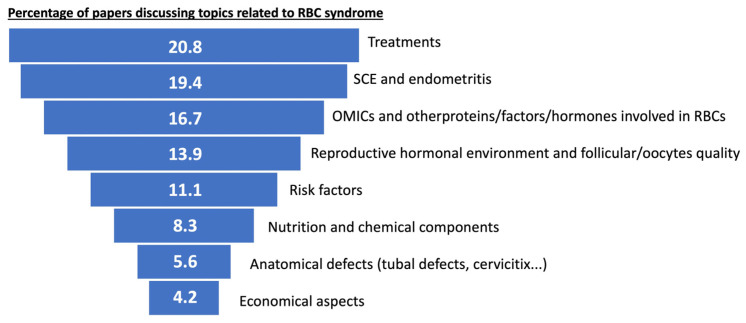
Subjects about RBC syndrome most frequently reported in the systematic review carried out between 2000–2023. The importance of SCE and other aspects related with the uterus and treatments used in RBCs were mainly studied by the researchers, but also new OMICs to determine the role of genes, proteins, or factors in the RBCs captured the interest of scientists.

**Figure 3 animals-13-02187-f003:**
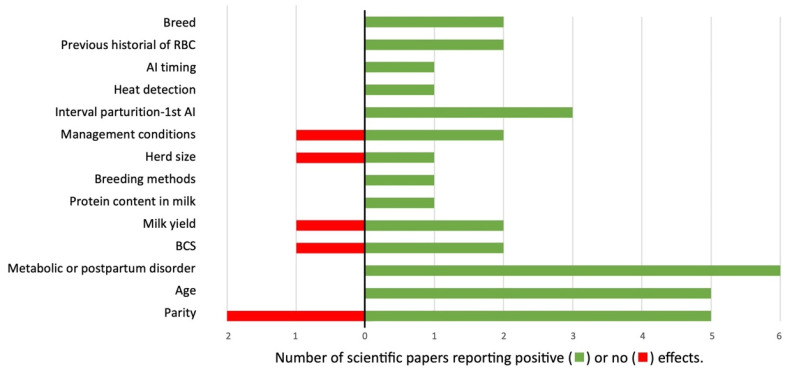
Impact of the risk factors studied on the incidence of RBC syndrome. Green bar indicates significant effect, while red bar informs that it does not affect the incidence of RBC syndrome. Values indicate number of scientific reports which observed effect or no effect of the evaluated risk factors.

**Table 1 animals-13-02187-t001:** Effects of different treatments implemented in RBCs on the pregnancy/conception rate. Data were collected from scientific papers between 2000 and 2023. Different letters (a, b, c) in each treatment indicate significant differences (*p* ≤ 0.05). ^1^ Pregnancy rate (PR); ^2^ Conception rate (CR).

Treatments						
	RBC		Control		*p* Value	Cites
	n	PR/CR	n	PR/CR		
Pg from d5 to d19 after AI	143	36.4 ^1^	148	33.8 ^1^	>0.05	96
CIDR from d10 to d19 + PFG2α on d18 +AI	30	43.3 b ^2^	30	20.0 c ^2^	<0.05	103
Similar + other CIDR from d5 to d13 after AI	30	63.3 a ^2^			<0.01	
GnRH before AI + Pg on d4, 5 and 6 + meloxican on d16, 17, and 18	98	37.7 ^2^	107	20.6 ^2^	<0.05	105
GnRH (single dose between day 7 and 14 after AI)	98	49.0 ^1^	90	37.8 ^1^	<0.001	120
Dephereline on d5 after AI 250 µg	271	39.1 a ^1^	269	28.6 b ^1^	<0.05	119
100 µg	270	39.1 a ^1^				
GnRH + 7-day P4-GnRH-PGF2α CL	191	70.0 ^1^	-	-		106
no CL	52	28.9 ^1^			<0.001	
GnRH + CIDR + PGF2α + EB + GnRH	498	32.0 ^1^	no control			107
GnRH 20 µg buserelin	55	87.0 a ^1^	42	48.0 b ^1^	<0.05	116
10 µg buserelin	40	58.0 b ^1^			>0.05	
1500 UI HCG on day 4	136	32.4 a,b ^1^	139	30.9 b ^1^	>0.05	112
on day 6	131	38.9 a ^1^			= 0.05	
Treatment 5–6 days after AI re-used CIDR	25	56.0 b ^1^	27	29.6 a ^1^		108
hCG	25	60.0 b ^1^				
GnRH	26	26.9 a ^1^				
Milk osteopontin 1.3 mg into vagina at AI	100	43.5 ^2^	100	22.2 ^2^	<0.05	42
Scopolamine 40 mg/100 kg on heat	20	80.0 ^2^	20	25.0 ^2^	<0.0001	125
Insulin on d8, 9, and 10 + PGF2α on d12	11	63.6 ^1^	10	40.0 ^1^	>0.05	81
Antibiotics Gentamycin one month before AI	10	80.0 a ^1^	14 (no RBC)	93.0 a ^1^	<0.05	124
Enrofloxacin one month before AI	12	33.0 b ^1^				
FTET in recipient RBC CL	208	32.2 ^1^	-	-	0.01	109
no CL	214	23.4 ^1^				
PGF2α-estrus	229	18.3 ^1^				
ET in RBCs also inseminated or not	IA and ET		ET			97
Heifers	61	49.2 b ^1^	61	29.5 a ^1^	<0.05	
Cows	273	41.5 b ^1^	137	20.4 a ^1^	<0.05	

**Table 2 animals-13-02187-t002:** Compilation of risk factors that could affect the incidence of RBC syndrome. Reviewed scientific reports used different statistical approaches to analyze the risk factors. Data are expressed as odd ratio (OR), percentage (%), or other not numerically expressed (asterisk indicates higher probability of RBC syndrome). Percentages have been estimated when studied groups are different than those indicated in the headline.

Risk Factor		Groups		Effect (Yes/No)	Authors
Parity	<2	3–4	>5		
	Ref.	3.3	4.8	yes	14
	9%	21%	13%	yes	128
	33.1% (1 parity)	29.1% (2–3 parities)	47.7% (>4 parities)	yes	7
	15%	9.8%	11.2%	yes	8
	Ref.		8.5 (>4 parities)	yes	129
				no	130
				no	126
**Age**	**<4**	**5–6**	**>6**		
	Ref.	1.7	7.6	yes	14
	14.4%	17.0%	14.1%	yes	128
	29.7%	33.1%	37.1%	yes	7
			*	yes	10
	Ref.	2.7	9.2	yes	14
	21.9%	12.5%	33.9%	yes	126
**BCS**	**Poor**	**Medium**	**Good**		
	9.7	Ref	1.1	yes	14
	42.3%	31.4%	26.3%	yes	7
				no	130
**Milk yield**	**<10**	**10–20**	**>20**		
	Ref	1.6	5.5	yes	14
	21.1% (1–2 l)	26.3% (2–5 l)	52.6% (>5 l)	yes	7
				no	130
**Herd size**	**<10**	**10–20**	**>20**		
	Ref.	1.1	2	yes	14
				no	130
**Breeding method**	**AI**	**Natural**	**Both**		
	4.1	1.4	Ref.	yes	14
**Metabolic disorder**	**Yes**	**No**			
	2.47	Ref.		yes	130
	*			yes	126
	*			yes	10
**Previous RBC condition**	**Yes**	**No**			
	*			yes	10
	*			yes	129
**Breed**	**Local**	**Others**			
		*		yes	128
		*		yes	17
**Interval calving-1st AI**	**<80 days**	**>80 days**			
	Ref.	0.78		yes	130
		*		yes	10
		*		yes	129
**Endometritis**	**Yes**	**No**			
	1.35	Ref.		yes	130
	*			yes	10
	*			yes	17
	2.0	Ref.		yes	14
**Sire**				no	141

## Data Availability

Not applicable.

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
