# Peer review of "Current Insights in the Repeat Breeder Cow Syndrome"

_animals, 2023, doi:10.3390/ani13132187_

Round 1
Reviewer 1 Report
The manuscript “Current insights in the Repeat Breder Cow Syndrom” by Perez-Marin and Quintela delivers a review about the recent knowledge regarding the multifactorial complex of the RBC. It was interesting for the reviewer to read this collection and summary of data. Special attention was given to pathogenesis, risk factors, medical approaches for treatments and finally eight (a-h) conclusions were listed.
First, the process of reference selection was described and lastly 72 studies were included for this manuscript. Many results are discussed and additionally summarized in 3 figures and 2 tables. The reviewer recognize that only references were searched and identified under the category “repeat breeder” (or similar). He is wondering if this kind of searching limits the horizon to get an adequate approach to this physiological phenomenon of RBC.
In general the results are informative, multiplex and interesting, especially under the view that the RBC is still an important problem in modern dairy herds. Overall, the paper is well and systematically written, starts with an appropriate title and demonstrate a clear editorial approach to this complex reproductive disorder.
The reviewer appreciates the enormous effort to collect and to analyze the different results very much. The complexity and the integration of different factors are well documented. The reviewer appreciates furthermore the attempt to summarize and to analyze different parameter from the literature and to direct it to some newer conclusions. However there are some essential remarks:
In total 72 studies were included for this review – on the other hand 80 % of the literature is excluded, because these references were published before the year 2000. This is quite understandable with the view of ”recent insights” but it ignores the basic work regarding this reproductive problem and it would be interesting to see the scientific shift in the discussion of this reproductive disorder problem. Some factors play key roles in this physiological complex between cow- oocyte/embryo –environment. However, they are not representative in the recent literature because newer issues show up, despite the original problem remains the same. Incongruously some older references are included. It is not a surprise that the majority of this review and the newer literature focuses on subclinical endometritis, new technologies, and therapeutic approaches. On the other hand other factors like the nutrition, BCS and milk yield are hardly visible. The reviewer understand the lack of information – only 8 % of the nutritional references are linked with RBC. (This is the problem of the strict/ limited searching. However, he expects a more critical logical dispute of the references to elevate this manuscript from a “level of formal collection of results” to a more “scientific résumé” about RBC and this should include a scientific evaluation of the references and their results. Perhaps this requires also the inclusion and evaluation of other scientific references.
In this direction, it is interesting to read, that the first and sixth conclusion mirrors nutrient deficits or the NEB, but it is discussed only superficially in chapter 6. Figure 3 is a very well designed scheme. It gives many information on a quick view. It shows also the immense influence of metabolism on RBC – but it is not mirrored in the paper.
The further most important risk factors are age and parity in this review. This is the results of unreflect mechanical listing of factors without a critical scientific approach.
In chapter 9, L501- 506 the immunodeficiency of elder cow (> 5 years) is discussed very superficially. The reviewer does not agree with the outcome from these old references (1986 and 2002) and from Ethiopia and has own different results. We have to respect our cows; we need more sustainability and have to prolong their productive lifetime by more welfare. To classify five or six year old cows (table 2) as elder animals (problem animals ??) need thinking that is more critical. This will improve our understanding of sustainable dairy management.
Some minor concerns:
Figure 1 documents the selectin process very well – however it is space occupying.
Table 2 looks interesting, displaying many results and there evaluation – however is not self-explanatory. Hard to understand. In comparison to Fig. 3 – what is the certain added value in such a complex reproductive disorder syndrome.
In chapter 10 the economic impact is discussed. The reviewer agrees with the outcome and knows the facts about the economic reality. However recent articles/ reviews should underline also the ethical issues. Other studies try to improve the productive life time of our cows despite the occurrence of reproductive disorders. An own ethical statement or critical dispute would improve the manuscript, despite the lack of such issues in the references in the last 23 years and before. ANIMAL is published to a wide and diverse readership. We have to adress such issues in an adequate manner.
Author Response
Authors (A) want to express our appreciation to the reviewers for their perceptive and constructive criticisms. All their suggestions and recommendations have been considered to improve the manuscript.
Following are listed the answers (in bold) to the comments and changes are showed in the texts (using changes control).
Reviewer 1
The manuscript “Current insights in the Repeat Breder Cow Syndrom” by Perez-Marin and Quintela delivers a review about the recent knowledge regarding the multifactorial complex of the RBC. It was interesting for the reviewer to read this collection and summary of data. Special attention was given to pathogenesis, risk factors, medical approaches for treatments and finally eight (a-h) conclusions were listed.
First, the process of reference selection was described and lastly 72 studies were included for this manuscript. Many results are discussed and additionally summarized in 3 figures and 2 tables. The reviewer recognize that only references were searched and identified under the category “repeat breeder” (or similar). He is wondering if this kind of searching limits the horizon to get an adequate approach to this physiological phenomenon of RBC.
A: The search was limited just to address the review on those studies conducted in RBCs, trying to avoid confounded studies about infertility or reproductive failure with other causality. Anyway, other studies linked with this subject but not conducted in RBCs have been also used (and cited) when we consider that they contribute to a better understanding. Perhaps, a more circumscribed review about only one aspect here analyzed could require a different searching approach; however, for the here conceived review we considered that it is the best way to compile the more relevant advances in this topic.
In general the results are informative, multiplex and interesting, especially under the view that the RBC is still an important problem in modern dairy herds. Overall, the paper is well and systematically written, starts with an appropriate title and demonstrate a clear editorial approach to this complex reproductive disorder.
The reviewer appreciates the enormous effort to collect and to analyze the different results very much. The complexity and the integration of different factors are well documented. The reviewer appreciates furthermore the attempt to summarize and to analyze different parameter from the literature and to direct it to some newer conclusions. However there are some essential remarks:
In total 72 studies were included for this review – on the other hand 80 % of the literature is excluded, because these references were published before the year 2000. This is quite understandable with the view of ”recent insights” but it ignores the basic work regarding this reproductive problem and it would be interesting to see the scientific shift in the discussion of this reproductive disorder problem.
A: As mentioned in the review, the more highlighted advances have been caried out about the discover of genes or proteins associated with the pathophysiology of this syndrome. Also, during the last years, it has been stated that the control of reproductive events (using different synchronization programs, time-fixed insemination, etc.) could be efficient to achieve the conception in this syndrome. And some information has been published about the importance of nutrition; while all researchers agree that nutrition is one of the main factors affecting reproductive failures, further studies are necessary to clarify how nutritional management could improve reproductive performance in RBCs. And also, in the last decades, assisted reproduction technologies (as OPU, ICSI, IVP) have been developed, and it could be expected new knowledge about the deficiencies or damages noted in gametes, embryos or follicular fluid in RBCs; science has advanced in this topic, but very slow.
Some factors play key roles in this physiological complex between cow- oocyte/embryo –environment. However, they are not representative in the recent literature because newer issues show up, despite the original problem remains the same. Incongruously some older references are included.
A: Some old references are included due in terms of their importance for understanding the problem or because not new advances have been reported.
It is not a surprise that the majority of this review and the newer literature focuses on subclinical endometritis, new technologies, and therapeutic approaches. On the other hand other factors like the nutrition, BCS and milk yield are hardly visible. The reviewer understand the lack of information – only 8 % of the nutritional references are linked with RBC. (This is the problem of the strict/ limited searching. However, he expects a more critical logical dispute of the references to elevate this manuscript from a “level of formal collection of results” to a more “scientific résumé” about RBC and this should include a scientific evaluation of the references and their results. Perhaps this requires also the inclusion and evaluation of other scientific references.
In this direction, it is interesting to read, that the first and sixth conclusion mirrors nutrient deficits or the NEB, but it is discussed only superficially in chapter 6. Figure 3 is a very well designed scheme. It gives many information on a quick view. It shows also the immense influence of metabolism on RBC – but it is not mirrored in the paper.
A: As reviewer says, more information should be discussed about nutrition. However, as before mentioned, it is difficult to find relevant new information. This sections have been enhanced.
The further most important risk factors are age and parity in this review. This is the results of unreflect mechanical listing of factors without a critical scientific approach.
A: In the present review, authors include all those studies analyzing risk factors in RBCs (see table) and we are sharing the collected information. We explain why these factors (not causes) could favor the appearance of RBCs, based on previous experiences. And age and parity are recurrently analyzed as a risk factor in RBCs. We have tried to explain better the link between the analysed risk factors and the probability to appearance of RBC.
In chapter 9, L501- 506 the immunodeficiency of elder cow (> 5 years) is discussed very superficially. The reviewer does not agree with the outcome from these old references (1986 and 2002) and from Ethiopia and has own different results. We have to respect our cows; we need more sustainability and have to prolong their productive lifetime by more welfare. To classify five or six year old cows (table 2) as elder animals (problem animals ??) need thinking that is more critical. This will improve our understanding of sustainable dairy management.
A: The importance of sustainability is mentioned in the last section "Economical aspects of RBC". Authors are in agreement with the reviewer's considerations about sustainability, welfare and lifespan in cows. It is interesting to discuss about this concern, but we are showing results from others. Obviously, we will discuss longer about it, but literature and previous trials and reports show us the reality at farm level. Should this fact be changed? Of course, but it i necessary not only suggest the good awareness and responsibility of farmers, but it is mandatory a pact or agreement of Governments to reduce economical losses at farmers (as it is made with banks, automotive sector, or others). Of course, it is necessary to contribute to generate opinion, but solutions are linking to public politics.
In reference to the age of cows, it is supported by literature that the lifespan of dairy cows at farm is around 5-6 years, but the natural longevity is around 20 years. Obviously, society do not accept now this kind of intensive production and the dairy production paradigm should change to support more sustainable systems and also a more respectful and empathic management with animals.
Some minor concerns:
Figure 1 documents the selectin process very well – however it is space occupying.
A: We agree with the reviewer, but it is important to show this process in this kind of paper (review). We encourage the reviewer to accept maintaining this table.
Table 2 looks interesting, displaying many results and there evaluation – however is not self-explanatory. Hard to understand. In comparison to Fig. 3 – what is the certain added value in such a complex reproductive disorder syndrome.
A: It has been better explained at different sections.
In chapter 10 the economic impact is discussed. The reviewer agrees with the outcome and knows the facts about the economic reality. However recent articles/ reviews should underline also the ethical issues. Other studies try to improve the productive life time of our cows despite the occurrence of reproductive disorders. An own ethical statement or critical dispute would improve the manuscript, despite the lack of such issues in the references in the last 23 years and before. ANIMAL is published to a wide and diverse readership. We have to adress such issues in an adequate manner.
A: Instead of the productive/economical aspect of the cows in both dairy or beef systems, it is important to consider longer productive timelife in cows in order to offer a higher welbeing to the animals and, of course, to obtain a more sustainable production systems. And these concepts about welfare and sustainability are being developed in the last years. This is one of the reasons because it is important to resolve (at least partially) those reproductive disorders that could stimulate the farmer to cull the animals in an early productive stage, without consider other options. A current trend is to consume meat from the named "retired" dairy cows, i.e. dairy cows that are retired from the farm and they graze pasture from months to years before to be sent to the slaughterhouse. It is fine for the cows (which can reduce their stressful dairy lifestyle), and for consumer (who will enjoy of a good quality meat). However, it is not easy for the farmers because they need pastures and/or facilities, or to identify those sellers interested in acquiring these cows and with a new channel to commercialize these animals. These new realities have been partially included in the present review, although some of these topics are not the aim in this review.

Reviewer 2 Report
This is an interesting review focusing on a serious problem of the modern dairy farming. The authors have adequately presented different aspects of the syndrome compiling a good body of the relevant bibliography. This reviewer believes that this manuscript can be published after minor modifications, and extensive language corrections.
Below is a list of points that should be addressed
1. Line (L) 13: replace attempts to conceive with inseminations
2. L20 and throughout the text: Please avoid starting a sentence with And
3. L21 estrous instead of estrus
4. L19-21 please add in the absence of any evident clinical signs
5. L43-46 please provide a more comprehensive and accurate definition of the syndrome.
6. L26 Replace and with while
7. L27 replace instead with despite
8. L29-30 please rephrase for clarity
9. L33 use restore fertility in place of solve infertility
10. L35 I think estrus is more appropriate keyword than heat
11. L42 delete those
12. L44 estrus signs but estrous cycles
13. L56-57 replace feed ration with maintenance
14. L58-63. This vast variation between different countries should be somehow commented
15. L67 replace conditions with systems
16. L67-70 In dairy or in beef cows?
17. L74 please make clear what do you mean by bull defects (subfertlile bull, subfertile semen at AI or what?)
18. L89 The current
19. L127 In this subheading (3) the criteria set by different researchers on the diagnosis of subclinical endometritis on the basis of PMNs must be presented.
20. L146 replace protective substance with a pregnancy supporting hormone
21. L151 expression,
22. 153 subclinical endometritis
23. L153-155 please provide a relevant reference
24. L151-160 This part is a bit confusing; it start with a question for healthy animals, but the following scenario refers to cows with subclinical endometritis
25. L177 what is the imaging technique Ultrasonography?
26. L177-180 please rephrase
27. L186 which steroid hormones?
28. L192 please provide reference
29. L207-211 This part must be moved to the treatment options
30. L215(>2mm) this does not denote volume
31. L252-258 please match the genes with the deviated expression with their functions
32. L259 please name these genes
33. L330-332 Please briefly explain how leptin is associated with RBC
34. L332 ow?
35. L364 – 368 This part must be rewritten as hormonal alterations are mixed with their structural and functional effects
36. L376-380 It must be stated here that delayed ovulation results in overmatured (aged) oocyte, which has reduced fertilizing capacity
37. L390-393 It is difficult for the reader to understand that this sentence refers to OPU followed by IVP
38. L392-397 was the AMH associated with the number of small follicles?
39. L398-405 Please disclose some of these parameters
40. L414-417 I am always wondering how wise is to transfer a valuable embryo in a subfertile recipient, for which the identification of the causative factor of subfertility, in most cases is speculative.
41. L431 Please provide what is the likelihood of pregnancy for RBCs and control cows
42. L435 Please replace alternative with recommended
43. L437-440 Please rephrase, because in the way it is written, it may perceived that it refers to the first AI of the future RBCs
44. L475 Are these cows high producing RBCs
45. L478 estrus
46. L508 alterations of what?
47. L 513-516 please rephrase for clarity
48. L517-542. There is a problem with definitions here. Heat stress can be considered as a syndrome since it affects general physiology, metabolism, morbidity, and fertility. Thus, if heat stress is listed among the causes of RBS, we contradict with the definition of RBC as only healthy cows are included. But heat stressed cows are febrile and panting -for example- so they cannot characterized as healthy.
49. L552 replace strong with higher
50. L577 replace semen with inseminations
51. L625 ‘In some cases’ Form personal experience, I would say In almost all cases
52. L625 is instead of can be
The manuscript needs extensive linguistic (spelling, typo, and grammar) corrections.
Author Response
Authors (A) want to express our appreciation to the reviewers for their perceptive and constructive criticisms. All their suggestions and recommendations have been considered to improve the manuscript.
Following are listed the answers (in bold) to the comments and changes are showed in the texts (using changes control).
Reviewer 2
This is an interesting review focusing on a serious problem of the modern dairy farming. The authors have adequately presented different aspects of the syndrome compiling a good body of the relevant bibliography. This reviewer believes that this manuscript can be published after minor modifications, and extensive language corrections.
Below is a list of points that should be addressed
- Line (L) 13: replace attempts to conceive with inseminations.
A: The definition of RBC is not exclusively associated to artificial insemination. RBCs fail to conceive after three or more attempts, which could be made by artificial inseminations or by natural mounts. However, when animals are monitored for AI (as usually occurs in dairy farms), it is easier to know how many times AI was tried, while those farms under extensive production or without strict reproductive monitorization are unable to identify this syndrome. After this explanation, we consider that it could be better to maintain the sentence without mention the way to obtain conception, i.e. AI or natural mount. Anyway, if reviewer considers that "insemination" can be undertood by the readers as either one (AI or mounts), then we agree to insert the modification.
- L20 and throughout the text: Please avoid starting a sentence with And. A: The manuscript has been reviewed and modified.
- L21 estrous instead of estrus. A: It has been corrected.
- L19-21 please add in the absence of any evident clinical signs. A: The sentence has been completed.
- L43-46 please provide a more comprehensive and accurate definition of the syndrome. A: The definition has been clarified: "For many years, researchers and technicians have focused on the Repeat Breeder Cow syndrome (RBC), in which cows exhibit estrus signs at apparently normal intervals (i.e., estrus cycles between 17 and 25 days) but repeatedly fail to be pregnant after at least three attempts, despite the absence of apparent anatomical abnormalities or infectious diseases [1–4]."
- L26 Replace and with while. A: It has been replaced.
- L27 replace instead with despite. A: It has been replaced.
- L29-30 please rephrase for clarity. A: It has been reviewed.
- L33 use restore fertility in place of solve infertility. A: It has been changed.
- L35 I think estrus is more appropriate keyword than heat. A: We agree with this consideration.
- L42 delete those. A: It has been removed.
- L44 estrus signs but estrous cycles. A: It has been corrected.
- L56-57 replace feed ration with maintenance. A: It has been modified.
- L58-63. This vast variation between different countries should be somehow commented. A: It has been commented.
- L67 replace conditions with systems. A: It has been modified.
- L67-70 In dairy or in beef cows? A: Beef cows offer more difficulties to obtain data since they do not need to come back twice or three times every day to the milking parlour. Then, in general, the monitorization in beef cows (mainly when animals are reared under extensive systems) is not so exhaustive as occur in dairy cows.
- L74 please make clear what do you mean by bull defects (subfertlile bull, subfertile semen at AI or what?). A: It was clarified.
- L89 The current. A: It has been replaced.
- L127 In this subheading (3) the criteria set by different researchers on the diagnosis of subclinical endometritis on the basis of PMNs must be presented. A: It has been corrected. "The literature established different thresholds of PMN for the diagnosis of SCE in cows (Waganer and col (2017-a review). Cytobrush and low-volume flushing are used for the diagnosis of SCE, but it is important to consider that the proportion of PMN decreases over the postpartum period. A fixed threshold of 18% PMN is established for samples taken between 20-33 days postpartum, 10% PMN at 34-47 days postpartum (Kasimanickam and col 2004) and 5% PMN for cows between 21-62 days postpartum (Madoz and col 2013). In contrast, for nulliparous heifers, the threshold for SCE is only 1% PMN (Pascottini and col, 1993). Overall, many studies support the use of 5% PMN as a diagnostic threshold for SCE. Cows exhibit a weak post-mating inflammatory response after breeding, which differs from other species (Wendt and col, 2006). It is hypothesized that a certain influx of PMN into the uterus immediately or within the first week after AI might be associated with physiological and beneficial effects on conception (Kaufmann and col, 2009, Drillich and col, 2011). "
- L146 replace protective substance with a pregnancy supporting hormone. A: It has been modified.
- L151 expression, A: Comma has been included.
- 153 subclinical endometritis. A: It has been modified.
- L153-155 please provide a relevant reference. A: A reference has been added.
- L151-160 This part is a bit confusing; it start with a question for healthy animals, but the following scenario refers to cows with subclinical endometritis. A: The sentences have been modified.
- L177 what is the imaging technique Ultrasonography? A: Yes, it has been simplified, clarified.
- L177-180 please rephrase. A: Sentence has been rephrased.
- L186 which steroid hormones? A: They have been included: "(as estradiol-17ß or progesterone)", as reported by different authors.
- L192 please provide reference. A: A reference was included.
- L207-211 This part must be moved to the treatment options. A: It has been moved.
- L215(>2mm) this does not denote volume. A: The presence of fluid into the uterus is evaluated by ultrasonography and it is stated that the measure (in mm.) informs about the content and the grade of endometritis.
- L252-258 please match the genes with the deviated expression with their functions
A: "Gene expression has also been evaluated to identify new etiological causes in RBC. Puglisi et al. (2013) described a total of nine genes expressed in cumulus-oocyte complexes (COCs) from RBCs, compared to control cows. In RBCs, lower expression of annexin A1 was observed, which affects the in vitro maturation of oocytes and upregulates the phospholipase A2 gene. There was also reduced expression of the lactoferrin gene, which is linked to lower in vitro fertilization rates and lower quality embryos. Under-expression of interferon-stimulated exonuclease 20 kDa was found in RBCs, compromising reproductive behavior and the response of the COC to estrogens. Moreover, the oxidized low density lipoprotein receptor 1 was under-expressed, which is associated with ovulation defects. The fatty acid desaturase 2 gene was over-expressed in RBCs, affecting the metabolism of polyunsaturated fatty acids as well as oocyte growth and differentiation. Glutathione S-transferase A2 was over-expressed, while glutathione S-transferase A4 genes were down-regulated, both of which are implicated in oxidative damage and oocyte viability. Similarly, the glutathione peroxidase 1 gene was up-regulated and associated with increased oxidative stress. Additionally, endothelin receptor type A was increased in the COC of RBCs, suggesting dysregulation of meiotic resumption and follicle rupture, among other actions [47]. "
- L259 please name these genes. A: Genes were named. "Using gene ontology, it has been determined that interferon-τ-stimulated genes (as ISG15, SLC2A1, SLC27 A6 and CXCL), but also peroxisome proliferator activated receptors, are involved in coding the trophectoderm cell proliferation and migration, and prompted conceptus elongation on day 16 [48]."
- L330-332 Please briefly explain how leptin is associated with RBC. A: It has been enhanced. "Leptin is a protein produced by adipocytes and modulated by insulin, glucocorticoids, and sexual steroids. It provides information to the brain regarding the available energy reserves, and it also inhibits the production of hypothalamic neuropeptide, a powerful appetite-stimulating factor. A study conducted during the postpartum period reported lower concentrations of leptin in RBCs compared to fertile cows [67]. The strong association between fertility and nutrition suggests that leptin may play a crucial role in modulating reproductive function, particularly during the postpartum period, when negative energy balance and low leptin levels coexist. The resumption of ovarian activity will depend on appropriate nutritional and health management during the postpartum period [Mann and col., 2005]. Additionally, low leptin levels can induce fasting behavior, resulting in reduced food consumption, and may lead to altered ovarian function due to a reduction of LH receptors and estrogen synthesis by granulosa cells [Odle and col., 2017]."
- L332 ow?. A: It has been corrected: "low".
- L364 – 368 This part must be rewritten as hormonal alterations are mixed with their structural and functional effects
A: "Elevated progesterone levels before ovulation in cows can lead to delayed, irregular, or absence of ovulation. Additionally, the presence of persistent follicles, characterized by prolonged and/or high estradiol levels, can contribute to the development of sub-functional CL. As a result, insufficient progesterone is produced, which may hinder the maintenance of pregnancy [4]. Moreover, ovarian cysts have been observed in RBCs, potentially as a result of prior hormonal or uterine disturbances [73]."
- L376-380 It must be stated here that delayed ovulation results in overmatured (aged) oocyte, which has reduced fertilizing capacity. A: It has been corrected. "Adequate progesterone levels during the preovulatory phase are also important [19]. The presence of suprabasal levels (usually resulting from partial luteolysis failure) can prolong the growth period of the ovulatory follicle and promote the overmaturation of the oocyte, thereby reducing its fertilizing capacity. Consequently, there may be observed asynchrony between AI and ovulation (which it is delayed or impeded), hormonal imbalances (elevated estradiol), formation of a subdued CL, and endometrial changes [71]."
- L390-393 It is difficult for the reader to understand that this sentence refers to OPU followed by IVP. A: It has been changed. "Following in vitro embryo production in RBCs, which involved ovum pick-up for collecting oocytes, no differences were observed in the number of follicles aspired per session, oocyte recovery rate, or cleavage rate, although the production of blastocysts was markedly reduced [73]."
- L392-397 was the AMH associated with the number of small follicles? A: The paragraph has been clarified. "More recently, a study describes a significantly higher percentage of RBCs when serum anti-Müllerian hormone (AMH) concentrations are lower [74]. This hormone is associated with the antral follicle count and informs about the size of the ovarian primordial follicle reserve pool [Tenley and col, 2019]. Therefore, the reduced fertility in cows with low AMH concentrations could be attributable to lower oocyte quality, reduced sensitivity of theca, granulosa, and luteal cells to FSH and LH, impaired embryonic developmental competence, and a relatively lower progesterone concentration during the estrous cycle. Interestingly, higher AMH levels have also been associated with suboptimal fertility, suggesting that a larger antral follicle count causes an imbalance in gonadotropin secretion, with greater LH and lower FSH secretions [74]. Monitoring AMH values is suggested to reduce future RBC incidence, as it is a moderately heritable trait, although further studies should be conducted."
- L398-405 Please disclose some of these parameters. A: It has been mentioned here, but also in the section "Nutrition".
- L414-417 I am always wondering how wise is to transfer a valuable embryo in a subfertile recipient, for which the identification of the causative factor of subfertility, in most cases is speculative. A: We agree with the reviewer's comment. However, it is one of the options described to overpass this syndrome. Obviously, those cows having irregular cycles or undetected deficiencies will have less opportunities to get pregnant in normal cycles, and then, it will be more and more difficult if ET is carried out. The cost of the ET technique should be also considered. "Anyway, it should be considered that not always the causative agent of subfertility in RBC is known, and then, the probability that the transferred embryo was successful is really uncertain. For this reason, so-called "therapeutic embryos", obtained by in vitro production from oocytes derived from slaughterhouse ovaries, are transferred to RBCs, avoiding the use of genetically valuable embryos, which are more expensive. "
- L431 Please provide what is the likelihood of pregnancy for RBCs and control cows. A: It has been clarified."Approximately 24.3% to 31.4% of RBCs will be pregnant within 210 days after calving [4, 8], but the likelihood of RBCs becoming pregnant after 300 days postpartum is extremely low, with only an additional 8% chance [8]."
- L435 Please replace alternative with recommended. A: It has been corrected.
- L437-440 Please rephrase, because in the way it is written, it may perceived that it refers to the first AI of the future RBCs. A: The sentence has been modified. "These treatments can improve and synchronize ovulatory events, such as promoting proper follicular growth, facilitating ovulation, and supporting corpus luteum (CL) formation. For these reasons, when used in RBCs, the mentioned treatments can improve the reproductive events, achieving conception rates of around 50–60% [82, 83], although further studies should be carried out. "
- L475 Are these cows high producing RBCs. A: Yes, it has been clarified.
- L478 estrus. A: It has been corrected.
- L508 alterations of what? A: It has been clarified. "In general, a higher incidence of RBC is attributed to older cows, not specifically due to age itself but rather derived from postpartum diseases as milk fewer, dystocia or retained placenta [10]."
- L 513-516 please rephrase for clarity. A: It has been modified."Recently, a trial conducted on beef RBCs revealed that cows at low parity with repeated cycles have a higher probability of failing to become pregnant again in subsequent parities in comparison to cows with 1 or 2 repeated cycles. Furthermore, cows that require 4 or more AIs to achieve pregnancy will have greater challenges in becoming pregnant in successive parities [108]."
- L517-542. There is a problem with definitions here. Heat stress can be considered as a syndrome since it affects general physiology, metabolism, morbidity, and fertility. Thus, if heat stress is listed among the causes of RBS, we contradict with the definition of RBC as only healthy cows are included. But heat stressed cows are febrile and panting -for example- so they cannot characterized as healthy. A: When we mention "heat stress", we are talking about extreme environmental conditions, as occur in Spain during summer season, and nowadays during a large part of the year as a consequence of the climatic change, noted as higher temperatures and longer dry periods. Literature affirms that cows exposed to extreme heat conditions can suffer heat stress, and fertility is commonly affected. Can heat stress or heat temperatures induce the estrous repetition? As fertility is affected, the percentage of RBC can be increased. It has been removed the paragraph about "stress", but it has been maintained and improved the paragraph about "environmental conditions".
- L552 replace strong with higher. A: It has been modified.
- L577 replace semen with inseminations. A: It has been modified.
- L625 ‘In some cases’ Form personal experience, I would say In almost all cases. A: It has been changed.
- L625 is instead of can be. A: It has been changed.

Round 2
Reviewer 1 Report
The reviewer appreciate the disputations with my concerns regarding the first version. I see an improvement of the manuscript. Nevertheless not all recommendations and hints are reworked. It is obvious that not all reproductive problems in dairy herds can be discussed, summarized and evaluated with the only use of references under the heading “RBC”. However I appreciate the diligence of this work.